# Food Parenting Practices among Parents with Overweight and Obesity: A Systematic Review

**DOI:** 10.3390/nu10121966

**Published:** 2018-12-12

**Authors:** Chloe Patel, Eleni Karasouli, Emma Shuttlewood, Caroline Meyer

**Affiliations:** 1Applied Psychology, International Digital Laboratory, Warwick Manufacturing Group, University of Warwick, Coventry CV4 7AL, UK; c.meyer@warwick.ac.uk; 2Warwick Clinical Trials Unit, Warwick Medical School, University of Warwick, Coventry CV4 7AL, UK; e.karasouli@warwick.ac.uk; 3Weight Management Services, Specialist Surgery, University Hospitals Coventry and Warwickshire NHS Trust, Coventry CV2 2DX, UK; Emma.Shuttlewood@uhcw.nhs.uk

**Keywords:** children, eating disorders, eating behavior, feeding practices, obesity

## Abstract

Given the links between parental obesity and eating psychopathology in their children, it is important to understand the mechanisms via which unhealthy relationships with eating are passed from generation to generation. The aim was to review research focusing on food-related parenting practices (FPPs) used by parents with overweight/obesity. Web of Science, PubMed and PsycINFO were searched. Studies that included a measure of FPPs were considered eligible and were required to have examined FPPs by parental weight status. Twenty studies were included. Single studies suggest differences between parents with healthy-weight vs. overweight/obesity with respect to; food accessibility, food availability and modelling. Multiple studies suggest that several parenting strategies do not differ according to parental weight status (child involvement, praise, use of food to control negative emotions, use of food-based threats and bribes, pressure, restriction, meal and snack routines, monitoring, and rules and limits). There was inconclusive evidence with respect to differences in parental control, encouragement and use of unstructured FPPs among parents with healthy-weight vs. overweight/obesity. The findings of this review imply some differences between parents with overweight/obesity and healthy-weight and the use of some food-related parenting practices, however, they should be interpreted with caution since research remains limited and is generally methodologically weak. The review highlights opportunities for further research, and suggests improvements to current measures of FPPs.

## 1. Introduction

A child is ten to twelve times more likely to have obesity when they have two parents with obesity when compared to having two parents with healthy weight [1,2]. In addition, children are developing obesity earlier [3], increasing the risk of developing adiposity-related conditions later in life including type II diabetes mellitus, cardiovascular diseases, sleep apnoea, problems with physical function, and some cancers [4,5,6,7]. Not only is parental obesity linked to obesity in their children, it has also been implicated in the aetiology of eating disorders (EDs), such as bulimia nervosa [8], binge-eating disorder (BED) [9], and anorexia nervosa [10]. For example, patients with anorexia nervosa have cited that living with a family member with obesity was one of the causes of the development of their ED [11]. 

Both obesity and eating disorders present in a significant proportion of young people. For instance, in 2016, 41 million infants and young children were overweight or obese globally [12]. In the UK, approximately one third of 2–15 year old children have overweight or obesity [13,14]. ED prevalence is also high, approximately five percent of children aged thirteen to eighteen will suffer from anorexia nervosa, bulimia, or binge eating disorder, with lifetime prevalence rates of 0.9%, 1.5%, and 3.5% among women, and 0.3%, 0.5%, and 2.0% among men [9]. BED is the most prevalent eating disorder associated with obesity among adults and adolescents [9,15] where the transmission of disordered eating has been illustrated in research. Parents with obesity, reporting binge-eating disorder (BED) behaviours, are significantly more likely to also report overeating, and binge-eating behaviours in their children than parents without BED behaviours [16]. Furthermore, children of mothers with overweight and obesity exhibit higher levels of emotional eating than children of healthy-weight mothers [17]. 

Research suggests that a child’s diet and preferences for food are usually influenced by food environments, including the eating behaviours of their parents [18,19]. This influence is strongest in early childhood, where parents act as gatekeepers and role models around food [20,21]. One important approach to tackling obesity in childhood and prevent the development of disordered eating behaviours is to understand and positively influence the modifiable determinants of healthy eating behaviours early in life [18,22]. Food parenting practices (FPPs) have been found to be one of the environmental factors associated with the development of overweight and obesity in childhood [22], and encompass the behaviours used by parents to influence their child’s behaviours, attitudes, or beliefs around food and eating [23]. FPPs are defined as active techniques or behaviours used by parents to influence a child’s food intake [24,25,26]. Although the relationship between FPPs, child weight and dietary intake is complex and bidirectional [27], one known predictor of children’s Body Mass Index (BMI)/weight is parental BMI [28,29,30]. This association can be attributed to genetic predisposition and environmental factors [31,32], including FPPs. Indeed, parents have a vital role in modelling food choices and shaping their children’s food preferences [33,34]. 

Due to recognised inconsistencies in the terminology and definitions on parents’ food-related behaviours, a working group of experts critically appraised the FPP literature and devised a content map to guide future research and to assist with study comparisons [23]. The appraisal resulted in three higher-order FPP constructs: coercive control, structure, and autonomy support/promotion [23]. Coercive control involves FFPs such as restriction, pressure to eat, threats and bribes, and use of food to control negative emotions [23]. Structure involves FPPs such as rules and limits around food, limiting/guiding food choices, monitoring, meal and snack routines, modelling, food availability, food accessibility, food preparation, and unstructured practices [23]. Autonomy support or promotion involves FPPs, such as nutrition education, child involvement, encouragement, praise, reasoning, and negotiation [23]. For the purpose of this review the FPP map was adopted to guide the description of results.

FPPs that support autonomy are non-directive, for example, encouraging balance and variety around food and providing nutritional education [35]. Such FPPs are believed to stimulate healthy food intake, and prevent consumption of unhealthy foods [25]. Conversely, coercive FPPs are directive, for example, pressuring a child to eat, restricting unhealthy or snack foods and use of food-based threats and bribes [36]. 

The latter type of FPPs, although well intended to prevent overeating [26], have been found to be associated with increased childhood weight and obesogenic eating behaviours, such as emotional eating and overeating [37]. For example, the use of food-based threats has been shown to affect BMI in adulthood [38]. This is because the reward status placed on the restricted food(s) increases the food’s affective value [39] and desirability [40], thus making them more likely to be eaten in excessive amounts [41]. Retrospective research conducted among adults indicates a heightened preference for foods that were restricted in childhood and higher levels of emotional overeating in adulthood [38,42], increasing the risk of binge-eating and bulimia [43,44]. 

Additionally, the use of food to control negative emotions is another coercive FPP that has been found to be associated with increased child BMI [45] and eating in the absence of hunger [46]. Adults recalling their own parents’ use of food to control their behaviours as a child via reward or punishment have also reported higher levels of binge-eating and dietary restraint [38]. Further, pressure to eat beyond satiety is detrimental to a child’s ability to acknowledge and react appropriately to hunger and fullness cues which in turn influences food intake [47]. Loth and colleagues identified that pressure to eat and food restriction were both significantly and positively associated with disordered eating among adolescent boys [48]. 

Extensive research has also shown that parents who are concerned with their own weight and eating behaviours are likely to exert coercive FPPs when feeding their children [49,50]. However, later in life, the use of such FPPs are associated with children’s less healthy eating behaviours, and disordered eating [48,51,52]. Studies such as these suggest that parents may, unknowingly, be promoting disordered eating and subsequent excessive weight gain in their child/ren via the use of unhelpful FPPs and eating behaviours [53]. Furthermore, since the risk of obesity is greater for children with one or more parents with obesity, identifying the particular FPPs used by parents with overweight/obesity could be helpful in informing the development of family based interventions.

In order to understand the determinants of FPPs, Birch and Davison’s model of multiple interactions proposes that there are numerous familial influences on the use of FPPs [54]. The influences described in the model are: parental weight status, parental eating behaviours, child weight status, and child eating behaviours [54]. Although the model does not acknowledge all the environmental factors associated with the development of childhood obesity [54], the model is appropriate for exploring the influences at the parental level, such as parental weight, on the use of FPPs. 

In summary, the FPPs currently being used by parents with overweight and obesity are yet to be identified despite parental BMI being associated with eating disorders and the strongest predictor of child weight/BMI. Therefore, the aim of this review is to systematically identify and review the types of parental FPPs used by parents with overweight and obesity (defined by a BMI ≥ 25.0 [55]). To aid cross-study comparisons, minimise conflicting findings and move towards consensus in measurement, the results are presented under Vaughn and Colleagues’ three higher-order food parenting constructs of the content map [23].

## 2. Methods 

### 2.1. Search Strategy

Potential studies were identified from three relevant electronic databases: Web of Science, PubMed, and PsycINFO. Published, peer-reviewed articles that examined FPPs were included. The reference lists of all relevant articles were hand-searched to further identify any additional studies that may have not been captured by the searches [56]. There was no limit placed on the publication date. Database searches were initially conducted in January 2017 and updated on 7 September 2018. Search strategies for each database can be found in Appendix A. 

### 2.2. Selection Criteria

The inclusion of studies was based on the PRISMA checklist’s PICOS (Participants; Interventions; Comparators; Outcome and Study design) taxonomy [57]. Participants: Studies were eligible if they were conducted with participants who identified themselves as parents, primary caregivers, or legal guardians. Participants had to have been grouped by BMI status or equivalent (e.g., healthy-weight, overweight, or obese). Studies focussing on infant feeding and studies including participants with medical conditions or disabilities that may influence FPPs and/or weight (e.g., Prada-Willi syndrome, Anorexia Nervosa, Binge Eating Disorder, Type I Diabetes Mellitus) were excluded. Interventions: Studies needed to have used a measure of FPPs, e.g., the Child Feeding Questionnaire. Comparators: Studies were eligible where there was a comparison group of parents with healthy-weight. Outcome: Studies needed to have considered a relationship between parental BMI and FPPs. Study design: Studies conducted quantitatively (cross-sectional, laboratory-based observation, longitudinal) were included. Peer reviewed studies that were written in English were considered eligible. Individual case studies, prospective and protocol articles were excluded. Studies involving FPP intervention or manipulation were also excluded as these studies do not capture naturalistic FPPs. Furthermore, participating in an intervention study can raise awareness of participants’ unhealthy behaviours [58].

### 2.3. Article Screening

The most recent studies identified from the search were published in 2018 and the oldest study was published in 1969. The titles and abstracts were screened for potential inclusion by one author (CP). A second reviewer (DM) also independently assessed each potential article for inclusion to determine whether it could be excluded on the basis of the inclusion/exclusion criteria. Disagreements were discussed and resolved by consensus [59]. A third reviewer (CM) was consulted where there was uncertainty. Full texts of potentially eligible studies were then screened by one reviewer (CP) and verified by the second (DM). 

### 2.4. Data Extraction and Synthesis

Data from each article were extracted and tabulated to present the study information. A data extraction form was developed according to the Centre for Reviews and Dissemination guidance [59]. The review and narrative synthesis was guided by the PRISMA statement for systematic reviews (http://prisma-statement.org/prismastatement/Checklist.aspx) (Appendix A) [57], and was registered on the PROSPERO database (CRD42018108891). A meta-analysis was not appropriate due to the heterogeneity between studies. 

### 2.5. Quality Assessment 

Articles were scored on their methodological quality, internal and external validity using the NICE quality appraisal for quantitative studies checklist [60]. It has been used in previous systematic reviews [61,62] and was adapted for the purposes of this review. The scoring for each criterion in the checklist ranged from ++ (when all or the majority of criteria were fulfilled), + (the criteria have been partially fulfilled), to - (few or none of the criteria have been fulfilled). Due to the limited number of studies revealed by the review, no publications were excluded from the review based on quality scoring. Study quality was also independently assessed by the second reviewer (DM) to examine possible risks of study bias, as suggested by Moher and colleagues [57]. Publication bias was not assessed due to heterogeneity among studies. Inter-rater reliability was in the acceptable range, intraclass correlation coefficient (ICC) = 0.87, and was assessed using a two-way mixed, consistency, average-measures ICC to examine the degree of agreement in study ratings between the two reviewers (CP and DM). 

## 3. Results

### 3.1. Summary of Included Studies

The initial search yielded 5599 abstracts (Figure 1). A proportion of articles (*n* = 197) were removed due to duplication, and 5402 abstracts were screened. The majority of abstracts (*n* = 5356) were excluded upon review as they did not meet the inclusion criteria. Forty-seven full-text articles were retrieved and read, however, a further twenty-seven were excluded from this review for the following reasons: not reporting FPPs by parental weight status (*n* = 10), no demographic data on the number/percentage of parents per BMI category (*n* = 11), the article presented the results of an intervention (*n* = 4), the sample included parents with healthy-weight only (*n* = 1), and measured perception of hunger (*n* = 1). One additional study was identified from a systematic review article [63] that was not identified in the search. Twenty studies were included in this review [64,65,66,67,68,69,70,71,72,73,74,75,76,77,78,79,80,81,82,83,84]. 

Apart from one study, 19 of the 20 included studies used widely-accepted BMI cut-offs for overweight and obesity (≥25). Lipowska and colleagues [75] used body-fat status measured by a body composition analyser and grouped parents into either overfat, healthy or underfat categories according to societal norms proposed by Gallagher and colleagues [84].

The oldest studies included in the review were published in 2001 [64,69] and the most recent studies were published in 2018 [75,79] (Table 1: Study results). Of the 20 relevant studies, 16 were cross-sectional [64,66,67,68,69,71,72,73,77,78,81,85], three were observational [74,76,82], and one longitudinal [70]. Research was conducted in the USA (*n* = 9), the UK (*n* = 2), Germany (*n* = 2), Turkey (*n* = 1), Australia (*n* = 1), Australia and New Zealand (*n* = 1), Brazil (*n* = 1), The Netherlands (*n* = 1), Poland (*n* = 1), and China (*n* = 1). Mothers comprised the participants in the majority of the studies (*n* = 13).

All 20 studies used nonclinical samples. The sample sizes varied where the largest sample was over 3000 parents [85], the smallest sample size was 20 mothers [74] (Table 1). FPPs were measured using questionnaires (*n* = 17), observations (*n* = 2), and a conjunction of both (*n* = 1). The questionnaires used in studies varied, however, the Child Feeding Questionnaire (CFQ) and CFQ subscales appeared to be used most frequently [66,67,68,69,70,73,74,77,85]. Other measures used to collect FPP data included the Pre-Schooler Feeding Questionnaire (PFQ) [64], the Chatoor Feeding Scale (CFS) [86], the Parental Feeding Style Questionnaire (PFSQ) [75,81], the Toddler Snack Food Feeding Questionnaire (TSFFQ) [67], the Comprehensive Feeding Practices Questionnaire (CFPQ) [35,79,80], the Caregiver’s Feeding Styles Questionnaire (CFSQ) [87], the Feeding Strategies Questionnaire (FSQ) [79], the Parenting Strategies for Eating and Activity Scale (PSEAS) [83], and the Meals in our Household (MioH) [79] measure.

### 3.2. Study Quality

Using the National Institute for Health and Care Excellence (NICE) rating system, four studies were rated as poor (-), fourteen were rated reasonable in quality (+), and two studies were rated good (++). The majority of research examined (14 studies) was rated as reasonable in quality (Table 1). This means that the criteria for internal and external validity were partially met to a standard whereby any criteria that were not fulfilled, would be unlikely to change the study conclusions [60]. Four studies were rated as poor in quality. This means that the design of the study contained sources of bias, such as little consideration for confounding variables [68,74], small sample sizes [74], and little or unclear information about the study sample [67,68,76]. 

### 3.3. FPP Results

A summary of the studies can be found in Table 2. 

#### 3.3.1. Coercive Control

The term “coercive control” is a distinct type of control that reflects parental attempts to dominate, pressure, or impose parental will on the child [88]. FPPs that are coercive have been described as parent-centred strategies with the aim to meet parental goals and desires [23]. Such FPPs that have been identified by the review are the following: 

##### Parental control

The measures that assessed parental control over their child’s eating were heterogeneous. This, in turn, revealed an inconclusive relationship between parental weight and use of parental control. There is some evidence to suggest that mothers with overweight/obesity have less control over their child’s intake and, therefore, their child has more control around their own intake of food [81]. Specifically, Wardle and colleagues [81] found that mothers with overweight/obesity reported significantly less control over their child’s food intake on the PFSQ when compared to mothers with healthy-weight. Similarly Haycraft and colleagues [71] found significantly higher reports of mothers with overweight/obesity giving their child more control around eating, as assessed by the CFPQ, in comparison to mothers with healthy-weight. In contrast, two cross-sectional studies reported no significant differences between parents with healthy-weight, overweight and obesity and CFQ control [77] and PSEAS control [83]. 

In one laboratory-based observational study, fathers with overweight demonstrated significantly more struggle for control (efforts by parent or child to control feeding) than fathers with healthy-weight and obesity [82]. The authors suggested that fathers with overweight attempt to try and control feeding due to concern about their child’s weight. This finding was not observed among the mothers in the sample. 

##### Using food to control negative emotions

Using food to control negative emotions [23] is a behaviour used by parents in response to their child’s emotional state [35,89], and is suggested to influence emotional eating in adulthood [90]. In the reviewed studies, the use of food to control negative emotions was measured using the PFSQ emotional feeding, e.g., “I give my child something to eat to make him feel better when he is upset” [81], the PFQ using food to calm a child, e.g., “Gave something to eat/drink if the child was upset” [64], the CFPQ emotion regulation, e.g., “Do you give this child something to eat/drink if s/he is upset even if you think s/he is not hungry?” [71] and by newly-developed questions, e.g., “Do you use foods to comfort your child?” [78]. 

There were five studies that reported no significant difference between parents with healthy-weight, overweight and obesity and the use of food to control negative emotions. Raaijmakers and colleagues [78] also reported no significant difference between use of food to control negative emotions and maternal healthy-weight, overweight, and obesity. However, this assessment was dichotomous, and consequently the frequency of the use of this FPP is unknown [78]. Another study reported that mothers with overweight/obesity use food to soothe their child significantly less than mothers with healthy-weight [72]. 

##### Threats and bribes

Five of the twenty identified studies explored the use of food-based threats and bribes. The majority of evidence identified appears to show no significant difference between parents with healthy-weight, overweight and obesity and the use of food-based threats and bribes in exchange for a favourable outcome (e.g., good behaviour from the child [37]), despite the varied measurement of this FPP. Wardle and colleagues [81] reported no significant differences between parents with healthy-weight, overweight, and obesity and PFSQ instrumental feeding. Haycraft and colleagues study also reported non-significant findings among maternal healthy-weight, overweight, and obesity using the CFPQ food as a reward subscale where their data was collected from a large sample of mothers with healthy-weight, overweight, and obesity in a community setting [71]. Two further studies also concluded that maternal weight had no significant effect on the use of food based threats and bribes [73,78]. In contrast, however, one study reported that the odds of mothers with obesity using CFPQ food as a reward was higher than compared to mothers with healthy-weight [80]. 

##### Discipline

One study examined the use of discipline among parents with their children via the PSEAS, which asks parents whether they discipline their child for unhealthy eating [83]. There were no significant differences between parents with healthy-weight, overweight and obese and the use of discipline for eating unhealthy foods [83]. 

##### Pressure to eat 

Pressure to eat is a controlling, directive feeding practice that aims to increase a child’s food intake [91]. There appears to be no difference between parents with healthy-weight, overweight, and obesity and pressuring a child to eat. No significant difference was found on the PFQ pushing the child to eat more [64], CFQ pressure to eat [66,68,69,73,74,77], PFSQ prompting/encouragement to eat [81] CFPQ pressure [71], and laboratory observational prompting a child to eat [76]. One study however, reported that parents with healthy-weight used significantly higher levels of CFQ pressure to eat when compared to parents with overweight and obesity, suggesting that parents with overweight/obesity use pressure to eat less [85]. Francis and colleagues [69] reported that pressure to eat by mothers with overweight/obesity was significantly predicted by daughters’ adiposity, and mothers’ concern for daughters’ weight. Pressure to eat by mothers with healthy-weight on the other hand was significantly predicted by mothers’ perception of daughters as underweight [69]. 

##### Restriction

Restriction involves controlling a child’s intake of unhealthy foods [91]. Parents might control a child’s intake with the intention to limit unhealthy foods or to decrease or maintain a child’s weight [35]. Ten identified studies included the assessment of restriction which used the CFQ and the CFPQ [66,68,69,70,71,73,74,77,80,85]. The evidence suggests that there is no difference between parents with healthy-weight, overweight, and obesity and the use of restrictive FPPs. Five studies found no significant difference in CFQ restriction [66,69,70,74,77] among mothers with healthy-weight, overweight, and obesity. Additionally, there was no significant difference between mothers with healthy-weight and overweight/obesity on CFPQ subscales: restriction for health and restriction for weight [71]. It has also been reported that the odds of mothers with obesity using CFPQ restriction for health were lower compared to mothers of healthy-weight [80]. 

Contrary to the aforementioned findings, two studies did report a significant difference in CFQ restriction between mothers, caregivers and parents with healthy-weight and overweight/obesity [68,85]. Francis and colleagues [69] conducted a five-year longitudinal study that reported among mothers with overweight/obesity, restriction could be significantly predicted by maternal concern for their daughters’ weight regardless of their daughters’ actual weight status, maternal perception of daughters as overweight, and maternal investment in weight and eating issues. 

One study combined multiple subscales from the CFPQ, FSQ, and the MioH [79] measure, and analysed the three overarching food parenting constructs outlined by Vaughn and colleagues [23]: coercive control, structure, and autonomy. Roberts and colleagues reported that there was no significant difference between parents with healthy-weight, overweight and obesity, and use of coercive FPPs [79].

#### 3.3.2. Structure

##### Meal and snack routines

Meal and snack routines are created by parents and includes the “location, timing, presence of family members, atmosphere or mood, and presence or absence of distractions during meals and snacks” [92] (p. 106). With regards to mealtime structure, the evidence remains inconclusive as this was explored in only one identified study [64]. Specifically, Baughcum and colleagues [64] included a domain in the PFQ that assessed structure during feeding interactions. This domain asks about whether the child watched television during meals, whether the child had a set mealtime and snack routine and whether the mother sat down with the child during mealtimes. A significantly lower degree of structure during mealtimes was reported by mothers with obesity than mothers without obesity [64]. 

Only one study examined mealtime atmosphere which reported no significant difference in dyadic reciprocity (affective engagement and quality of relatedness between mother and child), dyadic conflict (conflicts between mother and child over eating), talk and distraction during feeding (mother or child attempts to engage or control each other by talking or distracting), and maternal non-contingency (parental inability to interpret and respond to child cues) among mothers and fathers with healthy-weight, overweight and obesity [82]. More research is needed to examine meal and snack routines and parental BMI.

##### Monitoring

Parental monitoring involves the degree to which the parent keeps track of a child’s food consumption [36]. The small amount of evidence identified appears to suggest no difference between parents with healthy-weight, overweight and obesity and monitoring. Four studies found no significant difference in CFQ monitoring and CFPQ monitoring [66,68,71,73] and parent weight. Costa and colleagues [68] suggested that rather than parental weight, parental concern about their child’s weight, i.e., where the child is at risk of developing overweight or is already overweight, is related to parental monitoring of their child’s eating which questions the direction of this relationship. In contrast, another study using the PSEAS, reported that underweight and healthy-weight parents monitor their child’s diet significantly more than parents with overweight and obesity [83], suggesting that parents with overweight and obesity monitor their child’s diet less. 

##### Food accessibility

Food accessibility involves how easy or difficult it is for a child to access food independently or with assistance [23]. Access to such foods was assessed using the TFSSQ, and only one study used this measure [67]. Compared to mothers with obesity, mothers with healthy-weight and overweight recall previously allowing access to sweets and snack foods significantly less [67], suggesting that mothers with obesity allow access to sweets and snack foods more frequently than mothers with healthy-weight/overweight. In this particular study, mothers were asked to recall their previous and current FPPs. The recollection of CFPs may have, however, been influenced by mothers’ current CFPs or weight status and therefore this non-significant finding should be interpreted with caution. 

##### Rules and limits

Parents may set rules and limits to clarify what, how much, when and where their child/ren should eat [23]. Rules around snack foods was assessed in two studies via the TFSSQ [67] and PSEAS [83]. There was no significant difference between mothers with obesity and without obesity regarding their implemented rules around snack foods (TFSSQ), however, this did approach significance [67]. Also measured in this study was mothers’ flexibility around snack foods (TFSSQ), where there was also no significant difference between maternal BMI and this FPP [67]. Limit setting is assessed on the PSEAS, and asks parents about their use of boundaries around the consumption of unhealthy foods [83]. In this study there were no significant difference among parents with healthy-weight and overweight/obesity and limit setting [83]. 

##### Food availability

The types of food available and unavailable in the home is described as food availability [23]. Parental encouragement of balance and variety around food and the home food environment was assessed by one study [71]. This study utilized the CFPQ [35] where there were significantly lower reports of encouraging balance and variety among mothers with overweight/obesity in comparison to mothers with healthy-weight. Further, mothers with overweight/obesity reported having a significantly less healthy home food environment [71]. However, the sample in this study lacked heterogeneity as the majority were identified as white (76%). 

##### Modelling 

One study with a rather large sample (*n* = 437) explored maternal BMI and food modelling using the CFPQ [71]. Mothers with overweight/obesity reported significantly less modelling of healthy eating in comparison to mothers with healthy-weight [71].

##### Unstructured practices

FPPs that are “unstructured” involve the absence of parental control or structure around child eating, examples include meeting the child’s demands, allowing the child to make inappropriate food-related decisions, and providing little guidance or direction [23]. 

Child control of feeding interactions is a domain in the PFQ and the CFPQ and asks mothers whether they let their child choose their food from what is being served, whether mothers make something different if their child did not like what was being served, and whether mothers allowed their child to eat snacks whenever their child wanted [35,64]. Three studies explored child control around eating and reported contradictory findings. Specifically, Baughcum and colleagues reported no significant difference in PFQ child control around eating between mothers with obesity and mothers without obesity [64]. However, Haycraft and colleagues reported that mothers with overweight and obesity gave their child significantly more control around eating when compared to mothers with healthy-weight [71]. Russell and colleagues also reported that the odds of mothers with obesity allowing child control (CFPQ child control) is higher when compared to mothers with healthy-weight [80]. 

Age inappropriate feeding is a domain assessed by the PFQ and asks mothers to report; for example, if they gave the child a bottle during the day and whether they fed the child themselves if they did not eat enough [64]. Only one study found that mothers with obesity used significantly more age-inappropriate feeding in comparison to mothers without obesity. However, this difference was no longer significant after adjusting for family income [64]. 

One study combined multiple subscales from the CFPQ, FSQ, and the MioH [79], and analysed the three overarching food parenting constructs outlined by Vaughn and colleagues [23]: coercive control, structure and autonomy. Roberts and colleagues concluded that in comparison to parents with healthy-weight, parents with obesity use significantly less structure FPPs (there was no significant difference between parents with healthy-weight and overweight). 

#### 3.3.3. Autonomy Support/Promotion

##### Child involvement

There was no significant difference between mothers with healthy-weight, overweight, and obesity, and involving their child in planning and preparing meals and encouraging participation in food shopping. This is based on just a single study examining maternal BMI and involvement using the CFPQ [71].

##### Encouragement

In contrast to pressure to eat, whereby parents demand that their child eats more, encouragement involves parental use of positive, gentle, and supportive behaviours that are non-coercive [23]. Parental encouragement aims for children to build habits around healthy eating [23]. 

Two studies assessed parental encouragement using the PSFQ [75,81] which presented contradictory results. Lipowska and colleagues [75] reported that, among a Polish sample of parents, mothers with healthy body fat (body fat composition was measured rather than BMI) reported PSFQ encouragement FPPs significantly less than mothers with an overfat body status, suggesting that mothers with overfat use more encouraging FPPs than mothers with a healthy body fat status. Wardle and colleagues [81] on the other hand, reported that there are no significant differences in the PSFQ encouragement among mothers with healthy-weight, overweight, and obesity. 

##### Praise

Vaughn and colleagues define praise as a form of positive reinforcement where parents provide verbal feedback to the child [23]. One study assessed praise in the PSEAS which asks parents whether they use praise when their child eats healthy snacks [83], which found no significant differences between parental BMI and use of praise. 

One study combined multiple subscales from the CFPQ, FSQ, and the MioH [79], and analysed the three overarching food parenting constructs outlined by Vaughn and colleagues [23]: coercive control, structure, and autonomy. Roberts and colleagues reported that there was no significant differences between parents with healthy-weight, overweight and obesity, and use of autonomy support FPPs [79].

##### Nutrition education

Teaching about nutrition involves parents providing information and skills to their children to aid their decision making about the foods they eat, thus supporting the child’s autonomy since this information guides volition, and eating behaviours. One study included the assessment of teaching about nutrition using the subscale from the CFPQ [71], however, due to subscale reliability in the study was excluded from the analyses. More research is warranted to explore this FPP further. 

## 4. Discussion

The aim of this review was to systematically identify the types of food-related parenting practices used by parents with overweight/obesity in comparison to parents with healthy weight. This is important since extensive research indicates an increased presence of EDs among individuals who have parents with overweight and/or increased BMIs [9,93,94,95]. 

With regards to coercive food parenting practices, there is evidence (based on eleven studies) suggesting that there is no difference among parents with healthy-weight, overweight and obesity in their use of food to control negative emotions, use of food-based threats and bribes, pressure to eat and restriction [64,66,68,69,70,71,73,74,76,77,78,81]. The evidence examining parental control was inconclusive due to contradictory study findings [71,77,81,83]. These results are of interest as previous research suggests that parental weight status is a predictor of the use of coercive FPPs [54]. Parents who struggle with their own eating and weight are more likely to use coercive FPPs with their children [69,96,97] and adolescents [85]. However, the results in the current review, that there appears to be no difference between parental weight and use of coercive FPPs, suggests otherwise. The use of such FPPs could rather, be more driven by other parental cognitions such as concern about their child’s weight rather than their own weight. This was evident in one of the identified studies that reported restriction and pressure to eat was significantly predicted by maternal concern for their daughters’ weight [69]. 

With regards to parenting practices involving structure, there also appears to be no difference between parents with healthy-weight, overweight and obesity and: meal and snack routines, monitoring, or rules and limits [64,66,67,68,71,73,82,83]. However, the available research indicates that there are significant differences between parents with overweight and obesity versus parents with healthy-weight with respect to food accessibility, food availability, and modelling [67,71]. The research suggests that parents with overweight and obesity have a less healthy home food environment and model healthy eating less than parents with healthy-weight. Such findings shed light on the types of food environments children may be exposed to in families with overweight and obesity, which is one of the determinants of child weight [98]. Access and availability of healthy foods alongside parental modelling are all important FPPs in developing children’s healthy eating behaviours. For example, parental modelling of fruit and vegetable intake has been found to be positively associated with children’s fruit and vegetable intake [99] and lower availability of high-fat foods and sweet snacks [100]. Further, access to healthy foods might reduce the need for parents to exert coercive FPPs such as restriction. It should however, be highlighted that apart from rules and limits which was assessed in two studies, the structure FPPs described above were all examined in single, unreplicated studies. With regards to unstructured FPPs, the evidence was inconclusive due to contradictory study results [64,71,79,80]. FPPs that are unstructured include the absence of parental control [23], while this is an important for the development of child autonomy, having too much freedom with food choices and eating in addition to a less healthy home food environment, could result in less healthier selections of foods. It is important that unstructured FPPs are further researched particularly as eating behaviours in childhood can be tracked into adulthood [101], which emphasises the importance of the development of healthy eating behaviours in early life. 

Finally the results examining autonomy support FPPs, indicated that there are no significant differences between parents with healthy-weight, overweight and obesity and child involvement and praise which is also based on single, unreplicated studies [71,83]. Encouragement was examined in two studies, however, due to contradictory results, the evidence is inconclusive [75,81]. Although there was little evidence identified on autonomy support FPPs, they should be the focus of further research, since they provide parents the opportunity to convey information about healthy eating, subsequently allowing the child to internalise healthy norms and make informed decisions through the fostering of their autonomy [102]. 

The findings from this review should be interpreted with caution, since some FPPs in relation to parental BMI were examined in single studies, particularly where the research involved structure and autonomy support FPPs. In addition, it is unknown whether the research indicating that there is no relationship between parental BMI and FPPs is due to a real effect, the absence of methodological rigour (only two studies received ++ in this review) or the use of inadequate measures to capture FPPs. There may be value in conducting a review of measures using the COSMIN (Consensus-based Standards for the selection of health Measurement Instruments) checklist to aid the selection of the most appropriate measure for the FPP research at hand [103].

The current review identified numerous and inconsistent measures that are available to measure FPPs. Although the CFQ was the most frequently used measure to capture self-reported FPPs, many more feeding practices have been identified [23]. The CFQ does not capture the wider range of FPPs, such as parental modelling and teaching about nutrition [35], and so it is possible that there were additional FPPs used by parents that were not captured. It has been suggested that the inconsistent results between parental BMI and FPPs may be due to other variables, for example, parents own weight concerns, child age, and child weight [79]. On the other hand, it is possible that some of the inconclusive findings described above between parental BMI and FPPs are due to a lack of well-defined concepts being measured [92], subsequently resulting in a number of FPP measures that include similar subscales, but assessing different behaviours [92]. For example, the CFQ’s restriction subscale covers items about regulating the child’s intake such as limiting the amount of sweets and high fat foods consumed [36] and items such as, “I offer my child her favourite foods in exchange for good behaviour”. However, this is an item that others measures (such as the CFPQ Food as a reward subscale [35] and PSFQ Instrumental feeding subscale [81]) regard as food-based threats and bribes to behave [35]. 

Often only the minimal stages are used to design measures rather than what is required for rigorous measure development [92]. For example, seventy-one FPP measures have been identified in another systematic review, however, just less than half of these involved clear identification and definition of concepts to be measured during the development stage [92]. For the review this was problematic since there were limitations when comparing and evaluating the relationships between parental weight and subsequent use of FPPs among the studies included in the review. One of the strengths of the current review, however, is that the study findings were grouped and guided by Vaughn and Colleagues’ FPPs content map [23] that will help researchers plan future studies. 

### Study Limitations and Future Research

Several limitations have been identified. The samples in some of the studies may have introduced bias to the data identified in the review. For instance, Kröller and Warschburger [73] recruited mothers from clinics where they were receiving psychoeducation about their weight. Thus, their conclusion that maternal weight does not influence the use of FPPs might have been due to the mothers’ newly-acquired knowledge about the potential relationship between the use of certain FPPs and their children’s weight [73]. Two studies also reported there are no particular FPPs shared among mothers with overweight/obesity [64,77], however, this may have been due to mothers being recruited from the Special Supplemental Nutrition Programme for Women, Infants and Children where they may have been more attuned to eating behaviours before participating. 

Participants were predominantly white across the studies, so the generalisability of findings is restricted to other ethnicities. Two of the identified studies are applicable to white mothers and their daughters only [69,70]. Future research should seek to include more diverse ethnic samples, particularly as South Asian and Black Afro-Caribbean parents have reported greater pressure to eat [104], higher levels of restrictive FPPs and lower levels of monitoring [105] in comparison to White British and White German parents. 

Furthermore, household income is another sociodemographic characteristic that has been extensively associated with weight status [106,107,108] and so it is important that future research endeavours to collect this information. A small number of studies identified in the review did not collect this data [71,74,76,81]. 

In addition to the inclusion of family characteristics, the current evidence could be strengthened by larger sample sizes in future studies. Although Stevens suggests that “power is not an issue” when there is sample of 100 or more [109], none of the included studies presented a power calculation. Therefore, the results of those studies that included less than 100 parents with healthy-weight, overweight and obesity suggesting that there is little or no difference in the use of FPPs between parents with healthy-weight, overweight or obesity may have been due to studies being insufficiently powered [70,74,76,83], resulting in different statistical outcomes. 

With regards to study design, the current review identified only one longitudinal study [70]. The majority of studies were cross-sectional, which is an appropriate design for capturing the prevalence of behaviours without the risk of losing participants to follow-up (e.g., in longitudinal studies) [110]. However, neither the causality nor long-term impact of specific FPPs on child weight can be determined in cross-sectional studies. More longitudinal studies are required to further explore the relationship between parental BMI, FPPs, and childhood weight and eating behaviours. 

More research is also needed to help determine inconclusive and limited findings. Future research aiming to develop or improve measures of FPPs should do so using the appropriate steps for questionnaire development. Additionally, the bidirectional relationships that exists between parental FPPs and child eating behaviours should also be explored that includes parental BMI. It is also important for research to acknowledge that other adult caregivers may be influential on a child’s diet and eating behaviours. Parents are not only influential on their children, but also react, respond, and modify their FPPs to children’s behaviours and own parental feeding goals [111].

Although infant feeding was outside of the current review’s scope, it would be interesting to explore whether there is any relationship between parental weight and pressure (e.g., encouraging bottle emptying) with infants whom are bottle-fed. One of the concerns with the encouragement of bottle emptying is the interference with the infant’s ability to self-regulate their intake, and in combination with the parent’s potential to be unresponsiveness to the infant’s cues of satiety, can lead to unhealthy FPPs used with their child later in life. For example, frequent encouragement of bottle emptying has been found to increase the likelihood of the use of pressure-related FPPs in later childhood [112]. 

## 5. Conclusions

In conclusion, the findings of the review showed that studies with an improved methodological quality is required. A better understanding is required around the potential influence parental BMI has on the use of FPPs which may contribute to the parent-child BMI and eating behaviour relationship, particularly as FPPs are deeply influential on children’s eating behaviours and relationships with food later in life. This could be achieved by replication and extending of existing research including more longitudinal research with repeated use of the same or improved measures to capture FPPs [23]. Despite the mixed findings in the review, it is important that healthcare professionals working in weight management address disordered eating if successful weight-loss is the desired outcome. Similarly, it is important that healthcare professionals working with patients with EDs address weight management. Although more research is required, there may also be value in incorporating education around creating healthier home food environments within family-based interventions delivering nutrition education. 

## Figures and Tables

**Figure 1 nutrients-10-01966-f001:**
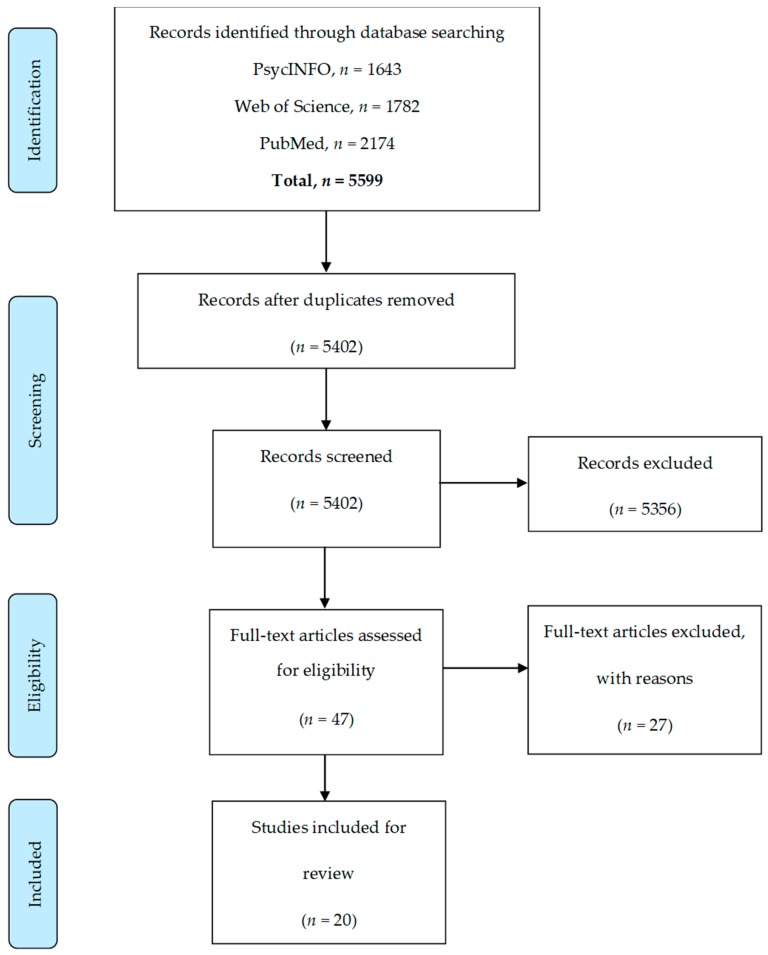
PRISMA Flowchart.

**Table 1 nutrients-10-01966-t001:** Quality rating by study.

Study	Quality Rating
Baughcum et al., 2001 [64]	+
Cebeci and Guven, 2015 [66]	+
Corsini et al., 2010 [67]	-
Costa et al., 2011 [68]	-
Francis et al., 2001 [69]	+
Francis and Birch, 2005 [70]	+
Haycraft, Karasouli, and Meyer, 2017 [71]	+
Jingxiong et al., 2008 [72]	+
Kröller and Warschburger, 2008 [73]	+
Lewis and Worobey, 2011 [74]	-
Lipowska et al., 2018 [75]	+
Lumeng and Burke, 2006 [76]	-
Powers et al., 2006 [77]	+
Raaijmakers et al., 2014 [78]	+
Roberts, Goodman, and Musher-Eizenmann, 2018 [79]	++
Russell et al., 2018 [80]	+
Wardle et al., 2002 [81]	+
Wendt et al., 2015 [82]	+
Williams et al., 2017 [83]	+
Berge et al., 2015 [85]	++

(-) indicates poor, (+) indicates reasonable in quality, (++) indicates good.

**Table 2 nutrients-10-01966-t002:** Study results.

Author(s), Study Country	Design	Aim(s)	Sample	BMI	*n*	Age of Children	FPP Measures	Relevant FPP Findings	Relevant Conclusions
**Baughcum et al. (2001), USA [64]**	Cross-sectional	To develop the Pre-schooler Feeding Questionnaire (PFQ).	634 mothers	18.5–29.9	488	23 months–5 year olds	PFQ	Significantly higher degree of age-inappropriate feeding (*p* = 0.004) (no longer true after adjusting for family income), concern about child overeating or being overweight (*p* = 0.001) regardless of child overweight and family income. Significantly lower degree of structure during feeding interactions (*p* = 0.001) (no longer true after adjusting for family income) among mothers with obesity vs. mothers without obesity. No significant differences on child control of feeding interactions (*p* = 0.070), using food to calm the child, concern about the child being underweight, difficulty in child feeding, and pushing the child to eat more (*p* values not reported) among mothers with obesity vs. mothers without obesity.	There is no specific feeding style associated with overweight young children.
				≥30	146				

**Cebeci and Guven (2014), Turkey [66]**	Cross-sectional	To examine the influence of maternal obesity on FPPs with their children with obesity.	491 mothers	18–24.9	41	6–18.5 year olds	Turkish CFQ	Other than perceived parent weight (*p* < 0.001), there were no significant differences in any CFQ subscales (concern over perceived responsibility (*p* = 0.494), perceived child weight (*p* = 0.093), concern over child’s weight (*p* = 0.152), restriction (*p* = 0.234), pressure to eat (*p* = 0.072), and monitoring (*p* = 0.782)) among mothers with obesity vs. mothers without obesity.	Maternal BMI does not appear to have a significant influence on FPPs.
				25–29.9	134				
				≥30	316				
**Corsini et al. (2010), Australia [67]**	Cross-sectional	To develop and validate the Toddler Snack Food Feeding Questionnaire (TSFFQ).	Sample 2: 216 mothers	≤18.5	2	4–5 year olds	TFSSQ and CFQ subscales: Restriction, Pressure to Eat and Monitoring	*Sample 2 (pre-schoolers, past practices)* Mothers without obesity allowed access to snack foods significantly less (*p* = 0.001), and implemented rules around snacking more (approaching statistical significance, *p* = 0.022) compared to mothers with obesity. No significant differences were found on any other constructs (*p* values not reported).	The TSFFQ is a useful measure that could be used in addition to other measures of parental feeding control.
				18.5–24.9	120				
				25–29.9	45				
				≥30	37				
				NR	12				
**Costa et al. (2011), Brazil [68]**	Cross-sectional	To examine parents feeding attitudes, parent BMI, and children’s weight status.	105 Parents/Care-givers	<25	68	6–10 year olds	Portuguese CFQ	Significant differences in perceived parent weight (*p* = 0.001), concern about child weight (*p* = 0.006), and restriction (*p* = 0.023) between parents with healthy-weight vs. parents with overweight/obesity. No significant differences in perceived responsibility (*p* = 0.861), perceived child weight (*p* = 0.844), pressure to eat (*p* = 0.233), and monitoring (*p* = 0.21) between parents with healthy-weight and parents with overweight/obesity.	Perceptions and attitudes of parents may independently be associated with overweight in children aged 6–10.
				>25	37				
**Francis et al. (2001), USA [69]**	Cross-sectional	To explore the predictors of the use of maternal restriction and pressure FPPs.	196 mothers	<25	92	5 year olds	CFQ subscales: Perceived child overweight, Child overweight, Restriction, and Pressure to Eat	There were no significant differences in the level of CFQ restriction and CFQ pressure to eat between mothers with healthy-weight and mothers with overweight. Among mothers with overweight, the use of restriction was significantly predicted by concern for daughters’ weight (*p* ≤ 0.05); pressure to eat was significantly predicted by daughters’ adiposity (*p* ≤ 0.05) and mothers’ concern for daughters’ weight (*p* ≤ 0.05).	Maternal weight status does not influence FPPs.
				≥25	104				
**Francis and Birch (2005), USA [70]**	Longitudinal	To explore restriction on food intake, the influence of eating in the absence of hunger on BMI, and maternal weight status as a mediator on these relationships.	171 mothers	≤24.9	80	5–9 year olds	CFQ subscale: Restriction	Overall, there was no significant difference in the amount of restriction used by mothers with overweight vs. mothers with healthy weight. Among mothers with overweight, use of restrictive FPPs significantly predicted daughters’ eating in the absence of hunger (*p* < 0.05).	There is no specific feeding style associated with mothers with overweight and obesity.
				≥25	91				
**Haycraft, Karasouli and Meyer (2017), UK [71]**	Cross-sectional	To compare maternal FPPs by maternal weight status.	437 mothers	19–24.9	249	2–6 year olds	CFPQ	Significantly higher reports of child control (*p* < 0.001) and lower reports of encouraging balance and variety (*p* = 0.029), environment (*p* = 0.021) and modelling (*p* < 0.001) among mothers with overweight/obesity vs. mothers with healthy-weight. There were no significant differences between mothers with healthy-weight/overweight and obesity on any other CFPQ subscales (involvement, monitoring, pressure to eat, restriction for health, restrictions for weight control, food as a reward, emotion regulation).	Mothers with overweight and obesity engage in fewer healthy FPPs when compared to a healthy weight sample of mothers.
				≥25	188				
**Jingxiong et al. (2008), China [72]**	Cross-sectional	To examine the relationship between FPPs and parental characteristics.	430 mothers	≤24	323	1–3 year olds	An interview to obtain information on: parent education level, family income, and FPPs (including a 24-h dietary recall)	In comparison to mothers with healthy-weight, mothers with overweight/obesity worry significantly more about their child overeating (*p* = 0.004) and that their child would develop obesity (*p* = 0.003). Significantly more mothers with overweight/obesity controlled feeding with a regular schedule in comparison to healthy-weight mothers (*p* = 0.017) and used food to soothe the child significantly less than healthy-weight mothers (*p* = 0.008).	Mothers with overweight report controlling child feeding with a regular feeding schedule and soothed children using food less often than mothers with healthy-weight.
				≥24	107				
**Kröller and Warschburger (2008), Germany [73]**	Cross-sectional	To explore the impact of various FPPs on child’s food intake and the influence of socio-economic status and weight on the use of different types of FPPs.	219 mothers	≤24.9	104	3–6 year olds	Items from the CFQ, CFSQ and newly developed questions from interviews with mothers and experts	No significant differences in FPPs between mothers with healthy weight and overweight/obesity. Maternal weight (underweight/healthy weight/overweight/obesity) had no significant effect on the use of FPPs (*p* = 0.60).	Maternal weight does not influence the use of FPPs.
				≥25	111				
**Lewis and Worobey (2011), USA [74]**	Laboratory observation	To explore maternal control and whether feeding style is different between healthy and overweight mothers.	20 mothers	<25	10	2 year olds	CFQ, food record, observed behaviours and video recordings.	No significant differences in pressure (*p* = 0.56) and restriction (*p* = 0.28), observed feeding style pressure (*p* = 0.49), and observed feeding style restriction (*p* = 0.28) between mothers with healthy weight and mothers with overweight/obesity. Mothers with overweight/obesity demonstrated significantly more concern about their own weight (*p* = 0.05) than mothers with healthy weight. Maternal BMI was not correlated with reported or observed feeding styles.	Lack of association between reported and observed feeding styles.
				≥25	10				
**Lipowska et al. (2018), Poland [75]**	Cross-sectional	To explore nutritional knowledge, eating habits, and appetite traits among children with and without excess body fat in the context of FPPs and body-fat status.	315 mothers; 276 fathers	Healthy * Mothers Fathers	190 109	5 year olds	PFSQ	Mothers with healthy body fat use encouragement to eat significantly less than mothers with an overfat body status (*p* < 0.05). Fathers with healthy body fat use control over eating significantly more than fathers with an overfat body status (*p* < 0.05). There were no significant findings on food as a reward and emotional feeding and parental body fat status (*p* values not reported).	Mothers with an overfat body status do not necessarily transmit unhealthy eating behaviours to their children.
				Overfat * Mothers Fathers	125 167				
**Lumeng and Burke (2006), USA [76]**	Laboratory observation	To explore if there is an association between maternal prompting to eat, child compliance and mother and child weight status.	71 mothers	<30	45	3–6 year olds	Parental prompting and child compliance	There was no significant difference found in prompting child to eat (*p* = 0.55) between mothers with and without obesity.	Greater maternal prompting was predicted by a younger child age, a novel food, more bites of food taken by the mother and low maternal education.
				≥30	26				
**Powers et al. (2006), USA [77]**	Cross-sectional	To explore the association of maternal feeding practices with maternal BMI and child eating behaviours.	290 mothers	<24.9	77	2–4.9 year olds	CFQ subscales: Restriction and Pressure to eat, PFSQ subscale: Control	There were no significant differences found with between maternal BMI and maternal FPPs: restriction (*p* = 0.63), pressure to eat (*p* = 0.33), and control (*p* = 0.62).	There is no particular feeding style shared among mothers with overweight or obesity.
				25–29.9	86				
				30–39.9	97				
				≥40.0	30				
**Raaijmakers et al. (2014), The Netherlands [78]**	Cross-sectional	To explore the use of instrumental and emotional feeding practices between main meals.	359 mothers	≤18.49	11	4–12 year olds	Self-constructed instrument developed from interviews with mothers and health promotion experts	Using food as a reward (26.8% of mothers with obesity) was reported more than use of food as a punishment (18.3% of mothers with obesity) and as a comfort (16.9% of mothers with obesity) with their child. No significant association between emotional and instrumental child feeding practices and maternal BMI.	Mothers offered energy dense and nutrient poor food items in emotional and instrumental child feeding practices.
				18.5–24.9	175				
				25–29.9	101				
				≥30	71				
				Over-weight (≥25)	5				
				Obese	10				
**Roberts, Goodman and Musher-Eizenmann (2018), USA [79]**	Cross-sectional	To investigate socioeconomic status, parental BMI and dieting status on the use of FPPs.	376 mothers; 118 fathers	18.5–24.9	223	2.5–7.5 year olds	CFPQ, FSQ, MioH, and newly developed questions	Post-hoc analysis revealed that in comparison to parents with healthy-weight and overweight, parents with obesity use significantly less structure FPPs. There was no significant difference between parents with healthy-weight and overweight. There was no significant post-hoc differences between parents with healthy-weight, overweight, and obesity and autonomy promotion (irrespective of a significant main effect) and coercive control.	When compared to other parental characteristics such as parental BMI, socioeconomic status has a small influence on the use of FPPs.
				25–29.9	149				
				≥ 30	120				
**Russell et al. (2018), Australia and New Zealand [80]**	Cross-sectional (secondary data analysis)	To explore FPPs among parents of toddlers and pre-schoolers and to examine the how FPPs differ by parent and child demographic data.	751 mothers	≤25	383	4–6 year olds	CFPQ	Among pre-schoolers (and adjusted for receiving a nutrition intervention before the measurement of FPPs), the odds of mothers with obesity using CFPQ food as a reward and CFPQ child control were higher compared to mothers with healthy-weight (OR = 1.13, 95% CI 0.94, 1.36; OR = 1.22, 95% CI 0.71, 2.09). The odds of mothers with obesity using CFPQ restriction for health and pressure to eat were lower compared to mothers with healthy-weight (OR = 0.86, 95% CI 0.72, 1.02; OR = 0.82, 95% CI 0.73, 0.91).	Nutrition interventions are unlikely to detect change in targeted FPPs since parents already report best practices, such as modelling and a healthy food environment.
				25 ≤ 30	186				
				≥30	152				
				NR	30				
**Wardle et al. (2001), UK [81]**	Cross-sectional	To identify any differences in feeding styles among mothers with obesity and normal weight.	Families with healthy-weight, over-weight and obesity	≤25	114	4–5 year olds	PFSQ	Mothers with obesity reported significantly less control over their children’s eating (*p* = 0.01) than mothers with healthy-weight. There were no significant differences in reports of emotional feeding, instrumental feeding, and prompting/encouragement to eat.	No difference in use of emotional, instrumental, and prompting/encouragement to eat parental feeding styles among mothers with healthy-weight, and obesity.
				Mothers ≥28.5 Fathers ≥25	100				
**Wendt et al. (2015), Germany [82]**	Laboratory observation	To explore parent-child interactions during feeding or joint eating and investigate the differences between mothers and fathers and parental weight.	148 mothers; 148 fathers	≤18.5 Mothers Fathers	4 2	7 months–3.9 year olds	Observation rated using the CFS	No significant differences found in CFS subscales: dyadic reciprocity, dyadic conflict, talk/distraction, struggle for control, and non-contingency among mothers with healthy-weight, overweight, and obesity. There were also no significant differences found among fathers with healthy-weight, overweight, and obesity apart from struggle for control. Fathers with overweight demonstrated a significantly higher amount of struggle for control than fathers with healthy-weight and obesity (*p* = 0.003).	Parents with healthy-weight, overweight, and obesity parents show the same ability to show relatedness, interpret child cues, and affective engagement during feeding and joint eating.
				18.5–24.9 Mothers Fathers	83 77				
				25–29.9 Mothers Fathers	17 32				
				≥30 Mothers Fathers	44 37				
**Williams et al. (2017), USA [83]**	Cross-sectional	To explore parental BMI and family behaviours associated with childhood obesity in a community sample.	143 parents	≤25	70	9–10 year olds	PSEAS	Underweight and healthy-weight parents monitor their child’s diet significantly more than parents with overweight and obesity (*p* < 0.000). There were no significant differences among parental BMI and discipline (children are disciplined for unhealthy eating), control, limit setting (boundaries with unhealthy eating), and reinforcement (praise for eating healthy foods).	Lower parental BMI is associated with a healthier home food environment.
				≥25	73				
**Berge et al. (2015), USA [85]**	Cross-sectional	To explore food restriction and pressure to eat by parent and adolescent weight concordance and discordance.	3252 parents	≤25	1444	Adoles-cents (mean age 14.4 years old)	CFQ subscales: Pressure to eat and Restriction	Parents with healthy-weight reported significantly higher levels of pressure to eat, compared to parents with overweight and obesity (*p* < 0.05). Parents with overweight/obesity reported significantly more food restriction compared to parent with healthy-weight (*p* < 0.05).	Use of FPPs are as a result of parental weight status and their adolescent’s weight status.
				≥25	2108				

* Determined using a segmental body composition monitor. Parental body fat percentage was calculated individually due to the differences in age. FPP: Food Parenting Practice, NR: Not Reported, BMI, Body Mass Index, PFQ: Pre-schooler Feeding Questionnaire, CFQ: Child Feeding Questionnaire, CFS: Chatoor Feeding Scale, PFSQ: Parental Feeding Style Questionnaire, TSFFQ: Toddler Snack Food Feeding Questionnaire, CFPQ: Comprehensive Feeding Practices Questionnaire, CFSQ: Caregiver’s Feeding Styles Questionnaire, PSEAS: Parenting Strategies for Eating and Activity Scale, FSQ: Feeding Strategies Questionnaire, MioH: Meals in our Household, OR: Odds Ratio, CI: Confidence Interval.

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
