# Peer review of "Food Parenting Practices among Parents with Overweight and Obesity: A Systematic Review"

_nutrients, 2018, doi:10.3390/nu10121966_

Reviewer 1 Report

Minor commnets:

- in the paragraph search strategy please add the date of last search of database;

- please cancel form the manuscript example search strategy and add all the search strategies as supplementary file 

- is there a possibility to fit table1 in the main file? it would be easier for the reader to find included studies

- after the sentence "Twenty studies were included in this review" should appear all the 20 references corresponding to the studies

Author Response

Response to Reviewer 1 Comments

We would like to thank reviewer one for their constructive comments on the submitted article. Please find responses to each comment below:

Point 1: In the paragraph search strategy please add the date of the last search of database.

Response 1: The date of database searches has been inserted on lines 130-131.

Point 2: Please cancel from the manuscript example search strategy and add all the search strategies as a supplementary file.

Response 2: The search strategies used in each database have been submitted with the revised manuscript as, “Supplementary file 1”.

Point 3: Is there a possibility to fit table 1 in the main file? It would be easier for the reader to find all included studies.

Response 3: The study results table has been inserted after line 447.

Point 4: After the sentence, “Twenty studies were included in this review” should appear all the 20 references corresponding to the studies.

Response 4: The references for all twenty studies has been included on line 189.

Reviewer 2 Report

The manuscript presented by Patel et al. is a systematic review that provides interesting information in relation to review research focusing on food-related parenting practices (FPPs) used by parents with overweight/obesity.

The paper is well written with a strong methodology by based on the PRISMA checklist’s PICOS (Participants; Interventions; Comparators; Outcome and Study design) taxonomy (http://prisma-statement.org/documents/PRISMA%202009%20checklist.pdf) and reporting of results.

But after careful analysis of the manuscript, I have a few general comments:

Abstract - I suggest to add information about study eligibility criteria.

Line 470-476: Section Study quality: maybe it would be better to present in the results section.

Make sure that the results highlighted in the results section are discussed in the discussion section as well. In my opinion, the discussion should be broadened (extend). The discussion needs several re-writing.

Line 487-488: Study limitations should consider other socio-economic factors, not just ethnicities group.

In addition, the impact of early feeding of the child (breastfeeding or bottle-feeding) should be considering. In literature are several papers investigate the role of maternal psychological parameters or weight in controlling behaviours related to the infant feeding (mainly bottle-fed), as well as the problem of "overfeeding" bottle-fed infants, which further interferes with proper development of the center of hunger and satiety.

It may be useful to include a few words in the conclusion regarding the implication of the findings for future practice and policy, not only by healthcare professionals.  For example, what could/should we be focusing on to strengthen nutrition education within parents/ children given the impact of this on food-related parenting practices (FPPs).

Make sure that all references are properly saved?

I also propose to format table S1 according to the journal’s instruction.

Instead of Annex 2 S2 - PRISMA checklist should be a link to the page
http://prisma-statement.org/prismastatement/Checklist.aspx - it is not an element of authors’ work?

Author Response

 Response to Reviewer 2 Comments

We would like to thank reviewer two for their constructive comments on the submitted article. Please find responses to each comment below:

Point 1: Abstract - I suggest to add information about study eligibility criteria.

Response 1: Adding all the eligibility criteria would compromise the abstract word count, therefore two of the main study eligibility criteria have been inserted in the abstract on lines 17-18.

Point 2: Line 470-476: Section Study quality: maybe it would be better to present in the results section.

Response 2: The paragraph relating to study quality has be moved and can now be found in the results section under section 3.2., lines 216 – 224. Additionally, a new table has been inserted (Table 1, line 234), presenting the quality rating by study.

Point 3: Make sure that the results highlighted in the results section are discussed in the discussion section as well. In my opinion, the discussion should be broadened (extend). The discussion needs several re-writing.

Response 3: The results have been further discussed and expanded on from line 458 onwards.  In addition, the whole discussion has been broadened as suggested.

Point 4: Line 487-488: Study limitations should consider other socio-economic factors, not just ethnicities group.

Response 4: Discussion of household income is included from lines 556-559.

Point 5: In addition, the impact of early feeding of the child (breastfeeding or bottle-feeding) should be considering. In literature are several papers investigate the role of maternal psychological parameters or weight in controlling behaviours related to the infant feeding (mainly bottle-fed), as well as the problem of "overfeeding" bottle-fed infants, which further interferes with proper development of the center of hunger and satiety.

Response 5: This is an important point and we thank the reviewer for their comment. We were keen to keep the focus of the review away from breast/bottle feeding since there has already been several reviews conducted (e.g., Appleton et al, 2018 (https://onlinelibrary.wiley.com/doi/abs/10.1111/mcn.12602); Norton et al, 2018 – (http://jccponline.com/Norton.pdf); Yeung et al, 2017 (https://www.liebertpub.com/doi/abs/10.1089/bfm.2016.0071)). Following much initial deliberation, we decided to focus the review specifically outside breast/bottle feeding, since we believe there is utility for the field in understanding more fully the later interactions. Whilst really important that further work is undertaken, and have added a short section into the discussion (lines 583-590) in response to this reviewer’s comment, we considered this area outside the scope of the current review.

Point 6: It may be useful to include a few words in the conclusion regarding the implication of the findings for future practice and policy, not only by healthcare professionals.  For example, what could/should we be focusing on to strengthen nutrition education within parents/ children given the impact of this on food-related parenting practices (FPPs).

Response 6: A short recommendation has been added in the conclusion on lines 601-603.

Point 7: Make sure that all references are properly saved

Response 7: These have been corrected.

Point 8: I also propose to format table S1 according to the journal’s instruction.

Response 8: As suggested by reviewer one, the study results table has been inserted into the article, and formatted as stipulated by the journal.

Point 9: Instead of Annex 2 S2 - PRISMA checklist should be a link to the page

http://prisma-statement.org/prismastatement/Checklist.aspx - it is not an element of authors’ work?

Response 9: Thank you for this comment. We have included the link as suggested on line 163. Authors submitting to Nutrients are advised to adhere to PRISMA guidelines, this includes submitting the PRISMA checklist and flow diagram. The flow diagram is included in the manuscript and the PRISMA checklist was submitted as a supplementary file. A recently published review in the journal’s special edition, “Eating disorders and Obesity: The Challenge for our Times” (Imperatori et al., 2018) also did this.